# Hsa_circ_0006692 Promotes Lung Cancer Progression via miR-205-5p/CDK19 Axis

**DOI:** 10.3390/genes13050846

**Published:** 2022-05-10

**Authors:** Jinrong Liao, Zeng Chen, Xingguan Luo, Ying Su, Tao Huang, Haipeng Xu, Keyu Lin, Qianlan Zheng, Lurong Zhang, Gen Lin, Xiandong Lin

**Affiliations:** 1Laboratory of Radiation Oncology and Radiobiology, Fujian Medical University Cancer Hospital, Fujian Cancer Hospital, Fuzhou 350014, China; jinrongliao@163.com (J.L.); czwyk99@163.com (Z.C.); zjsuying@126.com (Y.S.); lkyshy@sina.com (K.L.); zql15868162259@163.com (Q.Z.); lz8506@163.com (L.Z.); 2Beijing Huilongguan Hospital, Peking University Huilongguan School of Clinical Medicine, Beijing 100871, China; xingguang.luo@yale.edu; 3Shanghai Institute of Nutrition and Health, Chinese Academy of Sciences, Shanghai 200031, China; huangtao@sibs.ac.cn; 4Department of Oncology, Fujian Medical University Cancer Hospital, Fujian Cancer Hospital, Fuzhou 350014, China; penghaixu@163.com; 5Fujian Provincial Key Laboratory of Translational Cancer Medicine, Fuzhou 350014, China

**Keywords:** hsa_circ_0006692, mir-205-5p, CDK19, NSCLC, regulation of cancer malignancy

## Abstract

Circular RNA (CircRNA) is related to tumor development. Nevertheless, the regulation and function of hsa_circ_0006692 and its interactions with miR-205-5p and *CDK19* in the development of non-small-cell lung cancer (NSCLC) were un-explored. The correlations of expression levels of hsa_circ_0006692 in NSCLC specimens and cells with pathological characteristics were studied. The interactions of hsa_circ_0006692 with miR-205-5p and *CDK19* were assessed with real-time PCR, RNA-binding protein immunoprecipitation (RIP), luciferase reporter, RNA pull-down, and fluorescence in situ hybridization (FISH). The roles of hsa_circ_0006692 on cell growth, invasion, and migration in vitro and metastasis in vivo were evaluated. Hsa_circ_0006692 was over-expressed in 60 cases of NSCLC specimens and cells, which was positively correlated with TNM stage, tumor size, and invasion of the lung basal layer. The results of the in vitro and in vivo studies revealed that the over-expression of hsa_circ_0006692 facilitated NSCLC cell growth, migration, and invasion, cell cycle arrest at the S phase, and the activation of *BCL-2*, *CCND1*, and *PCNA*. The results of the dual-luciferase reporter assay, RNA immunoprecipitation, and pull-down assays indicated that hsa_circ_0006692 sponged miR-205-5p, which targeted *CDK19* and facilitated the malignant behaviors of lung cancer cells. Hsa_circ_0006692 modulated EMT of lung cancer cells via the stimulation of *CDH1*, *CDH2*, *VIMENTIN*, and *MMP7*. This study revealed that hsa_circ_0006692 promoted NSCLC progression via enhancing cell growth, invasion, and metastasis through sponging mir-205-5p, up-regulating the downstream oncogene *CDK19* and modulating EMT of lung cancer cells. The circ-0006692/mir-205-5p/*CDK19* axis might serve as a prognosis biomarker and target for drugs aimed against NSCLC.

## 1. Introduction

Lung cancer is the leading cause of tumor-associated mortality, with a 5-year survival ratio of merely 16.8% [1]. Among lung cancer, 85% are NSCLC [2]. There is an urgent need to identify the biomarkers and potential targets of NSCLC.

circRNAs, are a type of single-stranded closed RNA more stable than linear RNA; they play a critical role in gene regulation, including sponging microRNAs and regulating the translation of proteins [3,4]. Divergent primers (primers that cross each other back in primer pairs and reverse splicing sites) are employed to amplify specific circular transcripts [5]. The circRNAs engage in interactions with RNA binding protein and block the RNA from converting into proteins [6]. Pathologically, circRNAs are abnormally expressed in various cancers, such as cancers in the lung, gastric, liver, breast, colon, ovarian, skin, and so on. circRNAs are tightly associated with the clinicopathological features and prognosis of cancers [7,8,9,10]. For example, has_circ_0001649 expressing level in hepatoma is significantly different from that of normal liver expression and is strongly correlated with the cancer size and initiation of cancer thrombus [8]. The change of circRNA expression has a wide impact on the biological characteristics of lung cancer. For example, the circ_Pola2 promoted lung cancer cell stemness through the modulating miR-326/GNB1 axis [11]. CircRNA_102179 can promote the proliferative, metastatic, and invasive abilities of NSCLC cells through the modulation of the mir-330-5p/hmgb3 axis [12]. CircRNA hsa_circ_0008305 (circPTK2) suppresses TGF-β-triggered EMT and metastatic events through modulating TIF1γ in NSCLC [13].

However, the roles of hsa_circ_00006692 in lung cancer are unclear. Hsa_circ_00006692 is located on the 17:3608374-3608939chromosome, and its related gene symbol is *TCONS*. In the circExp database [14], the expression level of hsa_circ_00006692 is high in lung cancer on theGSE101684 dataset [15]. Our previous works have revealed that hsa_circ_00006692 was up-regulated in NSCLC compared to the adjacent samples. Herein, we explore if the abnormal expression of hsa_circ_0006692 can reinforce the malignancy of NSCLC. The results indicated that hsa_circ_0006692 could sponge miR-205-5p to inhibit its activity and up-regulate its target *CDK19*, affecting the proliferation and migration of NSCLC cells. Therefore, targeting the hsa_circ_00006692/mir-205-5p/CDK19 axis may be a new strategy for NSCLC treatment.

## 2. Materials and Methods

### 2.1. Ethical Statement and Specimens

A set of 60 paired human lung adenocarcinoma and paracancer specimens (over 5 cm away from the tumorous edge) were obtained from the Tumor Biobank of the Fujian Cancer Hospital, and the patients had under gone surgical treatment before any radiotherapy or chemotherapy at the Fujian Cancer Hospital between June 2018 and June 2020.The fresh tissue was frozen immediately at −80 °C until analysis.

This research was approved by the Ethical Board of Fujian Cancer Hospital (Fuzhou, China). All the patients involved gave written informed consent.

### 2.2. Cell Culture

Human lung cancer lineage cells (A549, H1299, BEAS-2B, HCC827, H358, PC-9) were provided by the Type Culture Collection of the CAS (Shanghai, China). All cell lines were validated via the STR method at the Shanghai Yihe Applied Biotechnology Co., Ltd. (Shanghai, China). The cells were cultivated with RPMI 1640 (Hyclone, Logan, UT, USA) medium supplemented with 10% FBS (Gibco, Grand Island, NE, USA), 100 μg/ml of streptomycin, and 100 IU/ml of penicillin in a 5% CO_2_, 37.5 °C cell culture incubator.

### 2.3. Hsa_circ_0006692 Over-Expression or Knockdown Cell Lines

For over-expression or down-regulation of circ-0006692 in cells, the viral vectors were made by the Hanheng Biological Technology Company (Beijing, China). For the cell line over-expression of hsa_circ_0006692, the hsa_circ_0006692 expression vector, HBLV-circ-0006692-Null-ZsGreen-PURO, was utilized. The control vector was the vector alone without any insert. For the knockdown of circ-0006692, three HBLV-circ-0006692-shRNA were utilized, and the parental vector, pHBLV-U6-MCS-CMV-Zsgreen, was the vector control. The cell infection and selection were carried out as previously described [16]. The newly established cells were named A549/NC, A549/circ-0006692-OE, H1299/NC, H1299/circ-0006692-OE, A549/circ-0006692-SH, and H1299/circ-0006692-SH, respectively. The symbols -OE or -SH represent over-expression and knockdown.

### 2.4. CDK19Over-Expression Cell Lines

A549/circ-0006692-SH cells cultured in 6-well plates (5 × 10^5^/well) were subjected to 10 MOI of HBLV-CDK19-Null-ZsGreen-PURO virus vectors infection with polyamine (6 μg/mL). Then, 48 h later, the cells were subjected to a selection with stylomycin (2 μg/mL) for 14 days to acquire the cells’ steady expression of *CDK19*, and it was named A549/circ-0006692-SH/CDK19-OE.

### 2.5. Establishment of A549 Cells Transient Over-Expression or Knockdown of miR-205-5p

Cells were transiently transfected with the miR-205-5p mimics or an inhibitor with X-tremeGENE HP DNA Transfection Reagent (Roche, UK). After 24 or 48 h, the cells were harvested for experiments.

### 2.6. Detection of mRNA Levels of Hsa_circ_0006692 and Proliferation Related Genes

The expression level of hsa_circ_0006692 mRNA was assessed in 60 paired lung cancer samples with qRT-PCR. The mRNA levels of the hsa_circ_0006692, *CCND1*, *PCNA,* and *BCL-2* in lung cancer cell lines (H1299 and A549) were also determined by qRT-PCR. The total RNA of the tissular specimens and lineage cells was extracted, and mRNAs were converted to cDNA via reverse transcription with M-MuLV reversed transcription enzyme (Promega, Madison, WI, USA) as per the supplier’s instructions. Real-time PCR was completed via SYBR1 Green Dye identification systems (Roche, Basel, Switzerland). The primers for magnifying the hsa_circ_0006692, CCND1, PCNA, and Bcl2 are presented in Appendix A. The comparative gene expressing levels were determined by the 2^−ΔΔCt^method. T37 was used as a reference, as reported previously [17,18]. For normalization, *GAPDH* was used as the endogenous control for hsa_circ_0006692, *BCL-2*, *CCND1*, and *PCNA*.

### 2.7. ColonyFormation Assay

The different infected cells (approximately 500 cells/well) were inoculated into 6-well dishes. After 2 weeks, the cells were washed twice in PBS and dyed in 0.2% gentian violet. Only the colonies with more than 50 cells were photographed with image scanning equipment (GE, Piscataway, NJ, USA). The number of colonies in all the wells was calculated via Image J.

### 2.8. Transwell Invasion Assay

The Transwell invasion experiment was conducted as described previously [19]. The invaded cells were pictured and then counted with Image J. The mean and standard deviation of cells in 5 regions per well were compared among the groups.

### 2.9. Wound Healing Experiment

The wound healing experiment was conducted as described in another study [19]. After 48 h, the images of the cells migrating into the empty line were captured using a microscope and analyzed using Image J.

### 2.10. MTS Assay and Cellular Cycle Analysis

MTS assay was carried out as previously described [19]. The absorption of all wells was measured at OD_490_ by Model 680 (Bio-Rad Lab, Hercules, CA, USA). Cellular cycle analysis was performed via the Muse™ cell cycle kit Muse™ Cell Analyzer (Millipore, Burlington, MA, USA), as per the supplier’s instructions [20].

### 2.11. Luciferase Assays

For the luciferase reporter assay, the 293T and A549 cells were seeded in 96-well plates. Negative control mimetic substances or miR-205-5p mimetic substances were co-transfected with the reporter plasmids into 293T and A549 cells via X-treme GENE HP DNA Transfectional Reagent (Roche, Switzerland. Cat No. 6366236001). Then, 48 h later, the cells were harvested, lysed with lysis buffer, and analyzed using a DR Tool (Promega, Madison, WI, USA) as per the supplier’s instructions. Renilla fluorescein enzyme activity was used as a normalized control.

### 2.12. Bioinformatics Analysis

Hsa_circ_0006692 sequencing information was acquired from circBase (http://www.circbase.org/, accessed on 04 May 2021). The targeted miRNAs of hsa_circ_0006692 were forecasted via starBase (http://starbase.sysu.edu.cn/index.php, accessed on 04 May 2021), BIOINF (http://www.bioinf.com.cn/, accessed on 04 May 2021), and circular RNA interactome (https://circinteractome.nia.nih.gov, accessed on 04 May 2021).

### 2.13. RNA FISH

Cy3, labeled hsa_circ_0006692 probe, was designed and prepared by the Ribobio Company (Guangzhou, China). The A549 cells were cultivated on round cover glass, fixed, infiltrated with PBS containing 0.5% Triton X-100, and subjected to dehydration in ethyl alcohol. FISH probes were desaturated and incubated in the cells overnight at 37 °C. After hybridization, the cells were dyed in DAPI for 10 min, and the slides were sealed with rubber adhesive and imaged with a fluorescence microscope.

### 2.14. RIP Experiment

RIP analysis was carried out with the EZ Magna RIP tool 17-701 kit (Merck millipore, Darmstadt, Germany). Briefly, A549 cells were harvested and scraped in RIP lysis buffer with protease and ribonuclease inhibitor. The cell lysates were added to antibodies against Ago2 or IgG (CST) and were treated overnight at 4°C. Then, the beads were washed and treated with protease K to realize the removal of the protein. The RNA was abstracted and studied using qRT-PCR.

### 2.15. Biotin-Coupled Probe Pull-Down Assay

RNA pull-down analysis was performed with a Magnetic RNA pull-down Tool (TFS). Biotinylated mir-205-5p mimetic substance (Bio-miR-205-5p) and NC (Bio-miR-NC) were provided by Ribo Biology (Guangzhou, China). A549 cells (10^6^) were subjected to a 100 nm biotinylation probe transfection for 48 h and then lysed with lysis buffer. Lysates were mixed with magnetic beads coated with streptavidin overnight. Then, the RNA sponged on these beads was separated, and the enrichment level of the circ-0006692 was determined via qRT-PCR.

### 2.16. Western Blot Analysis

Cell proteins were extracted with RIPA lysis buffer (KeyGen, Nanjing, China), and measured with the BCA kit (Thermo Scientific, Waltham, MA, USA) to determine their concentration. The protein mixture (30 μg) with the loading buffer (Takara, Kusatsu, Japan) was loaded onto SDS–polyacrylamide gel (10%; Invitrogen, Carlsbad, CA, USA), electrophoresed for 2h, transferred onto polyvinylidene fluoride membranes (Millipore, Billerica, MA, USA) and then blocked with WB Blocking Buffer (KeyGen, Nanjing, China) for 1h.The membranes were then incubated with the 1:1000 diluted primary antibody anti-light CDK19 (SAB, Cat No. 37480), MMP-7 (CST, Cat No. 3801S, CDH1 (CST, Cat No. 14472S), CDH2 (CST, Cat No. 13116S), VIMENTIN (CST, Cat No. 5741S), MMP-7 (CST, Cat No. 3801S), PCNA (CST, Cat No. D3H8P), CCND1 (CST, Cat No. E3P5S), BCL-2 (CST, Cat No. 5071S), and internal reference anti-β-actin (CST, Cat No. 4970), respectively, for 4 h, followed by 1:4000 diluted HRP-anti-rabbit IgG (CST, Cat No. 7074) for 1 h at 25 °C. Finally, the immune-conjugated signals were examined with Immobilon ECL Ultra Western HRP Matrix (Millipore, Cat No. WBULS0500), and the density of the bands was analyzed with the image station 4000 mm pro (Carestream, YTO, Toronto, ON, Canada).

### 2.17. Transplantation of Zebrafish with circ-0006692 Knockdown Lung Cancer Cells A549

Transgenic zebrafish TG (apo14:GFP) was provided by the Institute of Hydrobiology, Chinese Academy of Sciences. The transplantation of zebrafish experiment was performed following the protocol [16]. A549/circ-0006692-SH cells and A549/NC cells were injected with apicoliter injector. Under the microscope, 1500 cells were injected into the IVF developmental region of apo14: EGFP zebrafish yolk bag posterior for fertilization. After the injections, the embryos were cultured in a 33 ℃incubator and transferred into a 35 °C incubator after 24 h. The distribution and expression of the green fluorescent results from the zebra fish stomach cavity were pictured with laser-scanning confocal microscopic equipment (LSM 710, Karl Zeiss, Germany), and the expression level of the green fluorescent result in the zebrafish stomach cavity was determined with J 1.48v image.

### 2.18. Statistical Analysis

The entiredata were analyzed via spss19.0 (Chicago, IL, USA). Pearson correlation analyses were utilized. A two-tailed *t*-test was used to examine the diversity between the groups. A paired or non-parametric Kruskal–Wallis test was employed to assess the association between the hsa_circ_0006692 expression level and other features. *p* < 0.05 (* *p* < 0.05, ***p* < 0.01, *** *p* < 0.001 and **** *p* < 0.0001) indicated statistical significance.

## 3. Results

### 3.1. Characterization of Hsa_circ_0006692 in Lung Cancer and Its Association with Clinicopathological Features in NSCLC Patients

Hsa_circ_0006692 has 334 nucleotides and is derived from TCONS_00025531 in chr17 (3608375-3608939) (Figure 1A). Sanger sequencing was utilized to validate the qRT-PCR outcomes of the hsa_circ_0006692 expressed in the lung cancer samples. The results verified the head-to-tail splicing in the PCR products with the anticipated volume and forecasted splicing spot (Figure 1A). Then, the hsa_circ_0006692 was amplified in cDNA and gDNA using these two sets of primers_Circular and linear transcripts of circ_000669. The PCR results showed that the circular form of hsa_circ_0006692 was amplified by divergent primers in cDNA instead of gDNA, while the convergent primers amplified circ_000669 in both cDNA and gDNA (Figure 1C).

To identify the expression level of hsa_circ_0006692 inNSCLC, a qRT-PCR assay was performed in sixty tumor samples and their matched adjacent samples, one normal human lung epithelial cell BEAS-2B, and five NSCLC tumor lineage cells (HCC827, PC-9, H358, H1299, and A549). Figure 1D showed that the expression level of hsa_circ_0006692 in NSCLC (0.4731 ± 0.1984) was higher than adjacent tissues (−0.1608 ± 0.08829, *p* < 0.05). Similarly, the expression of hsa_circ_0006692 in the NSCLC tumor cell line was also much higher than in the normal lung epithelial cell line (*p* < 0.001, Figure 1E).

The clinical data revealed that the level of hsa_circ_0006692 expression was significantly related to clinicopathologic features, such as tumor size, TNM, and the invasion of the basal lung membrane in NSCLC patients (Table 1, *p* < 0.05, both). These results suggested that the expression level of hsa_circ_0006692 in NSCLC was significantly related to clinicopathologic features.

### 3.2. Overexpressed Hsa_circ_0006692 Promoted the Proliferation, Migration and Invasion of Lung Cancer Cells

To investigate the roles of hsa_circ_0006692 on NSCLC cells, the hsa_circ_0006692 over-expression lentivirus (HBLV-circ-0006692-Null-ZsGreen-PURO) or the control lentivirus were constructed and infected into A549 and H1299 cancer cells. The qRT-PCR results revealed that the expression level of hsa_circ_0006692 was remarkably elevated in hsa_circ_0006692 over-expression-infected cells (Figure 2A, *p* < 0.001). Functionally, the MTS assay and colony formation revealed that the proliferation of A549 and H1299 cells were 1.62 or1.55 times (MTS) and 1.32 or 1.67 times (colony formation) higher than in the hsa_circ_0006692 -OE group compared to the controls, respectively, *p* < 0.01, Figure 2B–E). The over-expression of hsa_circ_0006692 cells had an increased percentage of cells entered in the S stage (Figure 2F,G).Furthermore, the wound healing and transwell assays revealed that the migratory abilities and invasive abilities ofhsa_circ_0006692 over-expressed A549 and H1299 cells with increased 3.17, 1.51-fold (wound healing assay) and 2.34, 1.57-fold (transwell assay) compared with that in the controls, respectively, (*p* < 0.01, Figure 2L–Q). Moreover, the results of the RT-qPCR and WB revealed that the expression of the growth-associated genes *PCNA*, *CCND1*, and anti-apoptosis *BCL-2* were increased (Figure 2H–K).

### 3.3. Hsa_circ_0006692 Knockdown Inhibited Cell Growth, Invasive and Migratory Abilities of Lung Cancer Cells

To determine whether the knockdown of hsa_circ_0006692 could reverse its over-expression effects, the shRNAs for the junction site of hsa_circ_0006692 were infected into A549 and H1299 cells. The qRT-PCR results revealed that the expression of hsa_circ_0006692 was remarkably reduced in hsa_circ_0006692 shRNA-infected cells (Figure 3A). Functionally, the growth of hsa_circ_0006692 down-regulated A549 and H1299 cells decreased by 46.7%, 25.7% (MTS) and 28.2%, 54.9% (colony assay) compared to the controls (*p* < 0.001, Figure 3B–E), respectively (*p* < 0.001, Figure 3B–E). In lung cancer cells with the knockdown of hsa_circ_0006692, the proportion of cells arrested in the S stage was reduced (Figure 3F–G). In addition, the wound healing assay and transwell experiments revealed that hsa_circ_0006692 shRNAs remarkably repressed the migration to and invasion of lung cancer cells (Figure 3L–Q). The results of RT-qPCR and WB revealed that the expressions of growth-associated genes *CCND1*, *PCNA,* and anti-apoptosis *BCL-2* (Figure 3H–K) were reduced.

### 3.4. Hsa_circ_0006692 Acted as a “Molecule Sponge” for miR-205 to Promote the CDK19 Expression

The function of circRNAs is often related to different cellular localizations. FISH analysis revealed that hsa_circ_0006692 was primarily in the cytoplasm of lung cells (Figure 1B). To explore if hsa_circ_0006692 was capable of binding to microRNAs in the cytoplasm of A549 cells, a RIP analysis with an AGO2 antibody was carried out. As in Figure 4A, hsa_circ_0006692 could be magnified from the immune precipitate pulled down with the AGO2 antibody, revealing that hsa_circ_0006692 could bind to microRNAs through AGO2. Then, three databases (CircRNA interactome, BIOINF, and starBase) were used to search for the potential miRNAs, and hsa-miR-205-5p was found to be a binder with hsa_circ_0006692 (Figure 5A). To confirm these results, a pull-down test with a biotin-coupled hsa_miR-205-5p probe was performed. The results showed that the biotin-coupled hsa_miR-205-5p probe enriched hsa_circ_0006692 when compared to the negative control group (Figure 4B), suggesting that hsa_circ_0006692 was capable of directly interacting with miR-205-5p. The qRT-PCR results revealed that the over-expression of hsa_circ_0006692 suppressed hsa_miR-205-5p and increased *CDK19* expression (Figure 4C,E), while the knockdown of hsa_circ_0006692 increased the hsa_miR-205-5p and decreased *CDK19* expression levels (Figure 4D,E). The other three databases (TargetScan, PITA, and microRA.org) also indicated the binding of miR-205-5p with RNA of *CDK19* (Figure 5A).

To further evaluate the interactions of hsa_circ_0006692/hsa-miR-205-5p/CDK19 in lung cancer cells, we carried out a dual-luciferase report assay. We established the hsa_circ_0006692 wild-type luciferase reporter vector (pGL3-hsa_circ_0006692/CDK19-WT) involving the binding sequences of hsa_miR-205-5p and a mutant luciferase reporter vector (pMIRGLOhsa_circ_0006692/CDK19-Mut) where the mutation of the binding sequences occurred. The 293T and A549 cells were co-transfected with the WT or Mut fluorescein enzyme reporter and hsa_miR-205-5p mimetics. The results revealed that hsa_circ_0006692 and CDK19 WT fluorescein enzyme reporter activities were remarkably inhibited by hsa_miR-205-5p mimetics, whereas the activities of the Mut fluorescein enzyme reporter were not influenced by hsa_miR-205-5p mimetics (Figure 5B–D).

Whether hsa_miR-205-5p affected lung cancer cell activities triggered by hsa_circ_0006692 in vitro was examined with hsa_miR-205-5p over-expression or knockdown in A549 cells (Figure 4F). The qRT-PCR results revealed that knockdown of miR-205-5p improved hsa_circ_0006692 and *CDK19* expression levels (Figure 4H), while over-expression of miR-205-5p suppressed hsa_circ_0006692 and *CDK19* expression (Figure 4G).

These results revealed that hsa_miR-205-5p acted as a direct target of hsa_circ_0006692 and *CDK19* in lung cancer cells. The hsa_circ_0006692 is a “molecule sponge” for miR-205 to promote the expression of *CDK19*.

### 3.5. Hsa_circ_0006692 Plays Its Oncogene Roles through CDK19

To identify if hsa_circ_0006692 plays the role of an oncogene through interaction with *CDK19*, a rescue assay by promoting *CDK19* in A549/circ_0006692-SH was carried out. The results of qPCR and WB displayed that the expression of *CDK19* in A549/circ_0006692-SH/CDK19-OE was remarkably higher compared to A549/circ_0006692-SH (Figure 6A,B).

In addition, the *CDK19* over-expression could: (1) partially attenuate the promotion of cell proliferation (Figure 6C–D), invasion, and migration (Figure 6H–J) mediated by hsa_circ_0006692 knockdown; (2) partially attenuate the increased percentage of cell arrested in the S phase mediated by hsa_circ_0006692 knockdown (Figure 6E); (3) partially affected the hsa_circ_0006692-mediated regulation of EMT-related genes (Figure 6K); (4) partly rescue the up-regulation of proliferation-related genes mediated by hsa_circ_0006692 (Figure 6F,G). These data demonstrated that the hsa_circ_0006692 promotion of lung cancer proliferation, invasion, and migration acted partially through *CDK19*.

### 3.6. Hsa_circ_0006692 Facilitated the Progress of Lung Cancer in Zebrafish

To identify the biofunction of hsa_circ_0006692 in vivo, the zebrafish model was used. The results (Figure 7A,B) of the in vivo zebrafish assay with a stereomicroscope and confocal microscope at different time points to trace the fluorescence distribution of lung cancer cells with GFP (Green fluorescent protein) showed that the GFP-A549/circ-0006692-SH cells were 20% lower (*p* < 0.001) than that of the A549/NC in zebrafish at 72 h, indicating that circ-0006692 knockdown suppressed the proliferative ability of lung cancer cells in vivo as observed in vitro. In addition, the tail GFP area of A549/circ-0006692-SH was 8% lower than that of A549/NC at 72 h, indicating that the hsa_circ_0006692 knockdown inhibited the migration of lung cancer cells (*p* < 0.001, Figure 7C).

Those results revealed that hsa_circ_0006692 promoted lung cancer proliferation, migration, and invasiveness in vivo.

### 3.7. Hsa_circ_0006692 Modulated EMT of Lung Cancer Cells

EMT plays a critical role in cancer aggressiveness. Whether hsa_circ_0006692 could induce EMT in lung cancer cells and affect EMT-related genes was assessed via WB. As presented in Figure 2R,S, the over-expression of has_circ_0006692 promoted the expression of *CDH2*, *MMP7*, and *VIMENTIN*, while *CDH1* was down-regulated. Furthermore, the knockdown of hsa_circ_0006692 inhibited the expressions of *CDH2*, *MMP7*, and *VIMENTIN* (Figure 3R,S), while enhancing the expression of *CDH1* (Figure 3S). The rescue experiments also revealed that hsa_circ_0006692 expression was associated with EMT-related genes (Figure 6K). These results revealed that hsa_circ_0006692 could affect the activity of the EMT signaling pathway in lung cancer progression.

## 4. Discussion

Herein, the interactions and biofunctions of the hsa_circ_0006692/miR-205-5p/CDK19 axis in NSCLC progression were revealed for the first time as the following facts: (1) over-expression of hsa_circ_0006692 was related to tumor size, TNM, and the invasion of lung basal layer in NSCLC determined by60 paired NSCLC cancer and adjacent tissues; (2) hsa_circ_00006692 sponged miR-205-5p and inhibited its activity and then up-regulated *CDK19*; (3) hsa_circ_0006692/miR-205-5p/CDK19 axis influenced the proliferation, colony formation, migratory ability, invasion, and EMT of lung cancer cells; (4) hsa_circ_0006692 could modulate EMT of lung cancer cells. Collectively, these results illustrate that hsa_circ_0006692 is vital for the progress of lung cancer through the hsa_circ_0006692/miR-205-5p/CDK19 axis (Figure 7D).

CircRNAs are a huge classification of ncRNAs produced from back-splicing and have important modulatory roles in various cancers [21,22,23,24]. The present research supported the roles of hsa_circ_0006692’s in promoting NSCLC tumor progression via the enhancement of cell migration, invasiveness, and their expression of the proliferative ability with associated gene *PCNA*, *CCND1*, and anti-apoptosis *BCL-2*, EMT associated genes *CDH2*, *MMP7*, and *VIMENTIN*. The in vivo xenografts experiment with zebrafish also demonstrated the oncogene potential of hsa_circ_0006692. In conjunction with the higher hsa_circ_0006692 level identified in more invasive clinic NSCLC cases, this set of in vitro and in vivo findings consistently advocate for circ-0006692’s role in promoting the cell growth, migratory, and invasive abilities of lung cancer.

CircRNA is known to play a role in sponging miRNA via competitively binding to the appropriate miRNA, preventing miRNA from targeting mRNA to weaken miRNA-mediated gene suppression [25,26,27]. In cancer, such a sponge causal link was associated with cellular proliferation, migration, and angiogenesis [13,27,28]. The present study provided evidence to support the sponge effect of hsa_circ_0006692 on miR-205-5p in A549 cells: (i) endogenous circ-0006692 was enriched in the AGO2-FLAG IP fraction, revealing that hsa_circ_0006692 could be incorporated into RISC complexes (RNA-induced silencing complex); (ii) RNA pull-down assays revealed that hsa_circ_0006692 was enriched in the miR-205-5p biotin IP fraction; (iii) luciferase reporter assays confirmed hsa_circ_0006692’s binding to miR-205-5p; (iv) knockdown of miR-205-5p increased hsa_circ_0006692 and *CDK19* expression levels; and (v) over-expression of miR-205-5p suppressed hsa_circ_0006692 and *CDK19* expression levels.

MiR-205-5p is a well-known tumor suppressor miRNA species in many tumors. The down-regulation of miR-205-5p decreased the gemcitabine susceptibility of breast cancer cells through ERp29 up-regulation [29]; miR-205-5p contributed to the resistance to paclitaxel and the development of endometrial cancer via down-regulating *FOXO1* [30]; miR-205-5p/PTK7 axis participates in the growth, metastasis, and aggressionof colonic and rectal cancer cells [31]. In the present research, our team added the oncogene *CDK19* to the list of miR-205-5p genetic targets in lung cancer cells.

*CDK19* is a well-known oncogene in several cancers. Elevated mediator complex sub-type *CDK19* expression associated with aggressiveness of prostate cancer [32]; Pan-Cancer Analysis of the Mediator Complex Transcriptome determined *CDK19* and *CDK8* as treatment targets in late-period prostate carcinoma [33]. The impact of hsa_circ_0006692 KD (KnockDown) on promoting cellular proliferative, migratory, and invasive abilities in lung cancer cells could be reversed by the co-transfection of CDK19 OE (Over-Expression). These in vitro findings were verified by correlation analysis with clinical characteristics in patients with NSCLC. These findings suggest that hsa_circ_0006692 primarily exerts its promoting effect on NSCLC cell growth, invasion, and metastasis via sponging miR-205-5p and up-regulating *CDK19*.

EMT plays an important role in the course of the malignant development of cancer [34,35,36,37]. This study highlighted the importance of circRNAs in the regulation of the EMT. Forced CircPTK2 expression was discovered to suppress the metastasis of lung cancer cells and downregulate TIF1γ expression by altering the EMT process and decreasing *SNAIL* expression, suggesting that circPTK2 suppresses TGF-β-triggered EMT and metastases through the modulation of TIF1γ in NSCLC. Herein, Western blotting showed that overexpressed hsa_circ_0006692 resulted in the over-expression of *CDH2*, *MMP7*, *VIMENTIN*, and the down-regulation of *CDH1*, while the knockdown of hsa_circ_0006692suppressed the expression of *CDH2*, *MMP7*, *VIMENTIN* (Figure 3R,S), and the up-regulation of *CDH1* (Figure 3S). These findings revealed that hsa_circ_0006692 promoted EMT and malignance of NSCLC.

## 5. Conclusions

This study reveals that the hsa_circ_0006692/miR-205-5p/CDK19 axis plays a role in the growth, metastasis, progression, and EMT in NSCLC, which might serve as a prognostic biomarker or a new target for blocking the malignancy of NSCLC.

## Figures and Tables

**Figure 1 genes-13-00846-f001:**
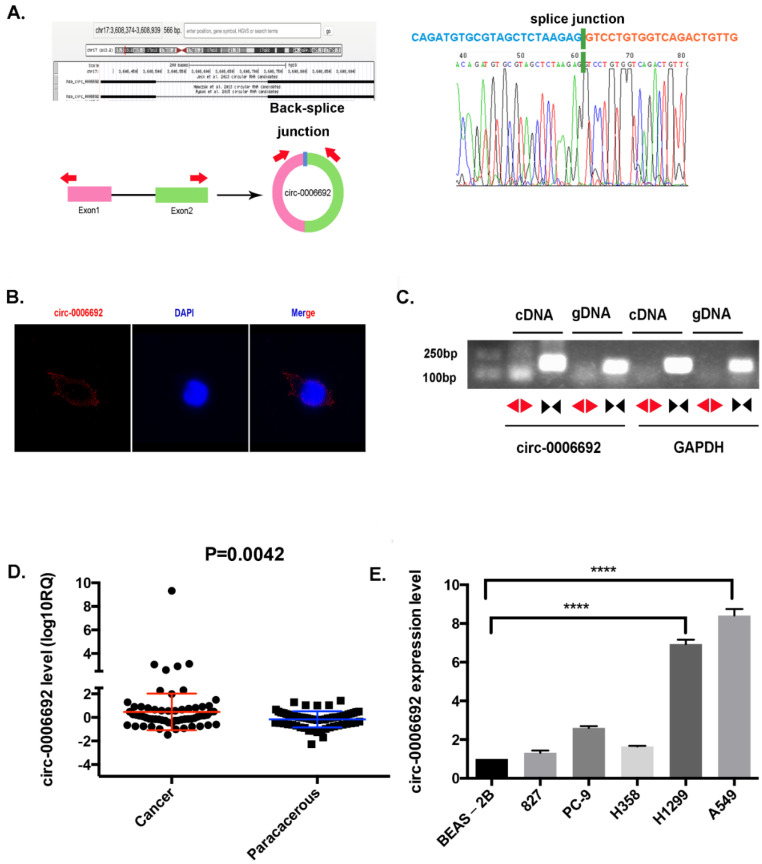
hsa_circ_0006692 is up-regulated in NSCLC. (**A**) hsa_ circ-0006692 locus sketch. The expression of circ-0006692 (circbase ID: hsa_circ_0006692) was verified by RT-PCR and Sanger sequencing. Red arrows represent divergent primers used to amplify the genomic region of circ-0006692 containing backsplice junction site (JCT); (**B**) Representative image of RNA fluorescence in situ hybridization for endogenous hsa_circ_0006692, shown localization of hsa_circ_0006692 in the cytoplasm of A549 cells. The cellular nucleus was subjected to counterstaining with 4,6-diamidino-2-phenylindole (DAPI). Scale bar, 5 μm; (**C**) In A549 cells, divergent primers amplified hsa_circ_0006692 in cDNA JCT, but not in genomic DNA (gDNA), and convergent primers amplify both hsa_circ_0006692 JCT and linear circ-0006692. GAPDH was used as a reference control. Red and black arrows represent divergent and convergent primers, respectively. (**D**) Expression of hsa_circ_0006692 in 60 paired lung cancer and adjacent tissues; (**E**) Expression of hsa_circ_0006692 in five human lung carcinoma cell lines (HCC827, PC-9, H358, H1299, and A549) and one normal human lung epithelial cell line (BEAS-2B). **** *p* < 0.0001.

**Figure 2 genes-13-00846-f002:**
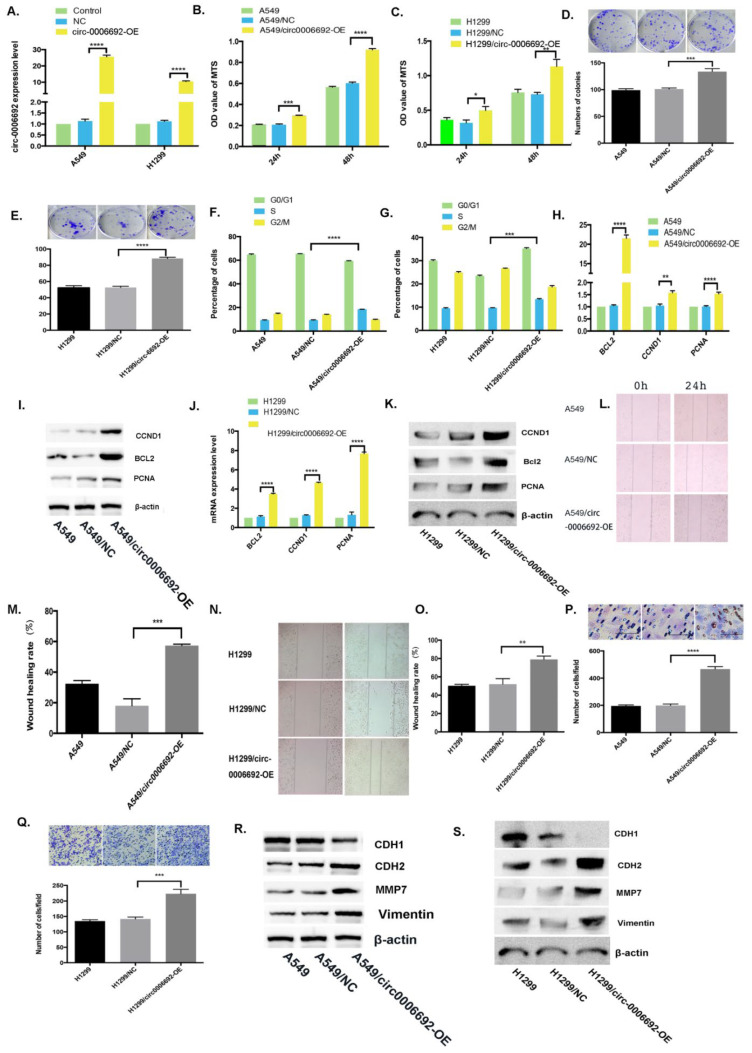
Over-expression of hsa_circ_0006692 promoted the proliferation, migration, and invasion of lung cancer cells. (**A**) qPCR on hsa_circ_0006692 expression in A549 and H1299 cells; (**B**,**C**) The roles of overexpressed hsa_circ_0006692 in the vitality of A549 and H1299 cells measured using the MTS assay; (**D**,**E**)The roles of overexpressed hsa_circ_0006692 in the colony formation of A549 and H1299 cells; (**F**,**G**) The role of overexpressed hsa_circ_0006692 in cell cycle distribution of A549 and H1299 cells measured with FACS; (**L**–**O**) The role of overexpressed hsa_circ_0006692 in the migration of A549 and H1299 cells measured by the wound healing assay; (**P**,**Q**) The role of overexpressed hsa_circ_0006692 in the invasiveness of A549 and H1299 cells measured using the transwell invasive assay; (**H**–**K**) The expressions of proliferation-related-genes of *Bcl-2*, *CCND1,* and *PCNA* were detected by qRT-PCR and WB; (**R**,**S**) The expression levels of invasion/metastasis-related-genes, *MMP7*, *VIMENTIN*, *CDH1*, and *CDH2* demonstrated by WB. * *p* < 0.05; ** *p* < 0.01; *** *p* < 0.001; **** *p* < 0.0001.

**Figure 3 genes-13-00846-f003:**
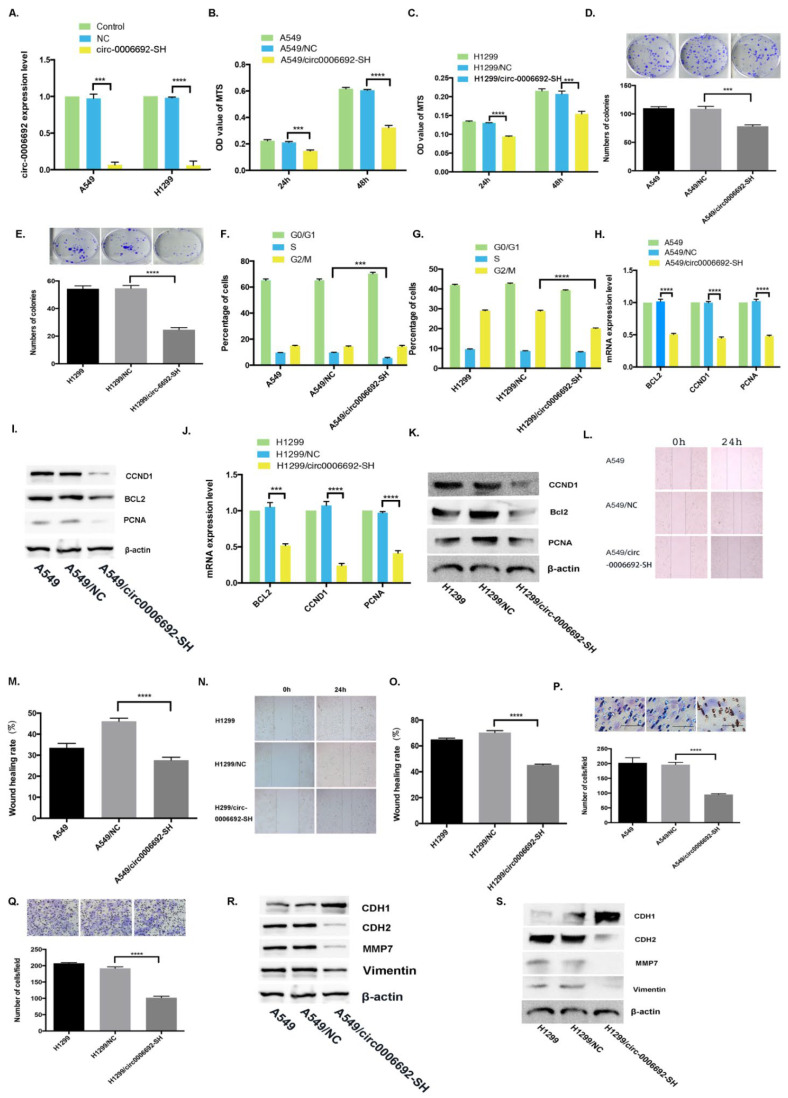
Silence of hsa_circ_0006692 inhibited the proliferation, migration, and invasiveness of A549 and H1299 cells. (**A**) qPCR on hsa_circ_0006692 expression in A549 and H1299 cells. (**B**,**C**) The roles of hsa_circ_0006692 silencing in the vitality of A549 and H1299 cells measured using the MTS assay; (**D**,**E**) The roles of hsa_circ_0006692 silencing in the colony formation of A549 and H1299 cells; (**F**,**G**) The roles of hsa_circ_0006692 knockdown in cell cycle distribution of A549 and H1299 cells measured with FACS; (**L**–**O**) The roles of hsa_circ_0006692 silencing in the migration of A549 and H1299 cells measured with wound healing assay; (**P**,**Q**) The roles of hsa_circ_0006692 silencing in the invasiveness of A549 and H1299 cells measured with the transwell invasive assay; (**H**–**K**) The expression of proliferation-related-genes *BCL-2*, *CCND1*, and *PCNA* were detected by qRT-PCR and WB; (**R**,**S**) The expression of EMT-associated genes, *MMP7*, *VIMENTIN*, *CDH1*, and *CDH2* demonstrated by WB. *** *p* < 0.001; **** *p* < 0.0001.

**Figure 4 genes-13-00846-f004:**
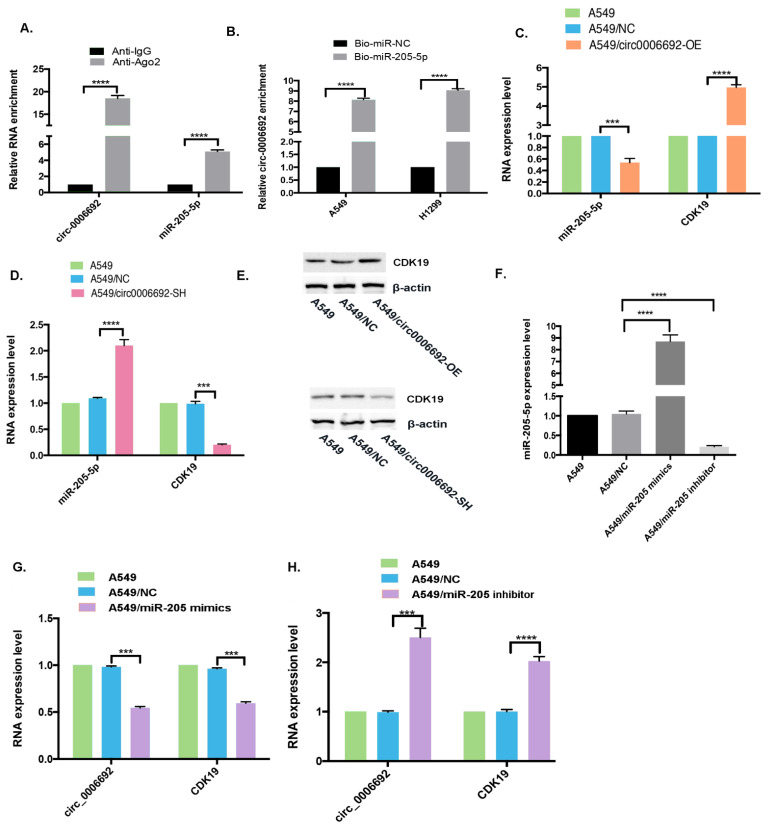
hsa_circ_0006692 played its roles via sponging hsa-miR-205-5p and regulation of CDK19. (**A**) After the detection of ago2 or IgG RIP assay, hsa_circ_0006692 and miR-205-5p levels was determined by qRT-PCR; (**B**) After RNA pull-down analysis using Bio-miR-NC and Bio-miR-205-5p in A549 and H1299 cells, expression of hsa_circ_0006692 was measured by qRT-PCR; (**C**–**E**) CDK19 expression level was increased in A549/circ-0006692-OE cells and decreased in A549/circ-0006692-SH cells analyzed by qRT-PCR and WB; (**C**,**D**) miR-205-5p expression was decreased in A549/circ-0006692-OE cells and increased in A549/circ-0006692-SH cells analyzed by qRT-PCR; (**F**) miR-205-5p expression in A549 cells transfected with miR-205-5p mimics or inhibitors was analyzed using qRT-PCR; (**G**,**H**) qRT-PCR analysis of hsa_circ_0006692 and *CDK19* expression in A549 cells transferred with miR-205-5p mimics or inhibitor. *** *p* < 0.001; **** *p* < 0.0001.

**Figure 5 genes-13-00846-f005:**
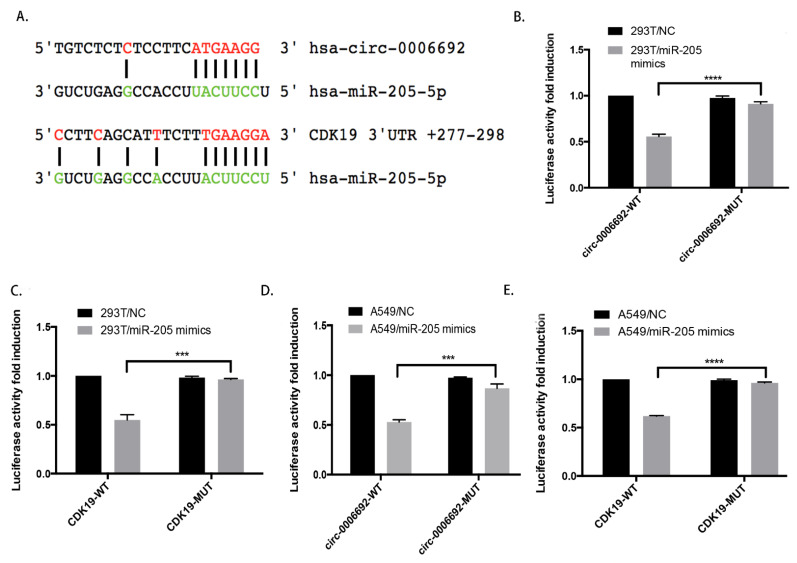
hsa_circ_0006692 played its roles via sponging hsa-miR-205-5p and regulating *CDK19*. (**A**) Sequence alignment of hsa-mir-205-5p with putative binding sites in circ-0006692 and CDK19 wild-type and mutant regions; (**B**–**E**) Double Luciferase Report analysis showed that hsa-mir-205-5p mimics reduced the fluorescence intensity of HEK293T or A549 cells transfected with circ-0006692-Wt or CDK19-Wt rather than circ-0006692- Mut or CDK19-Mut vector; ****p* < 0.001; **** *p* < 0.0001.

**Figure 6 genes-13-00846-f006:**
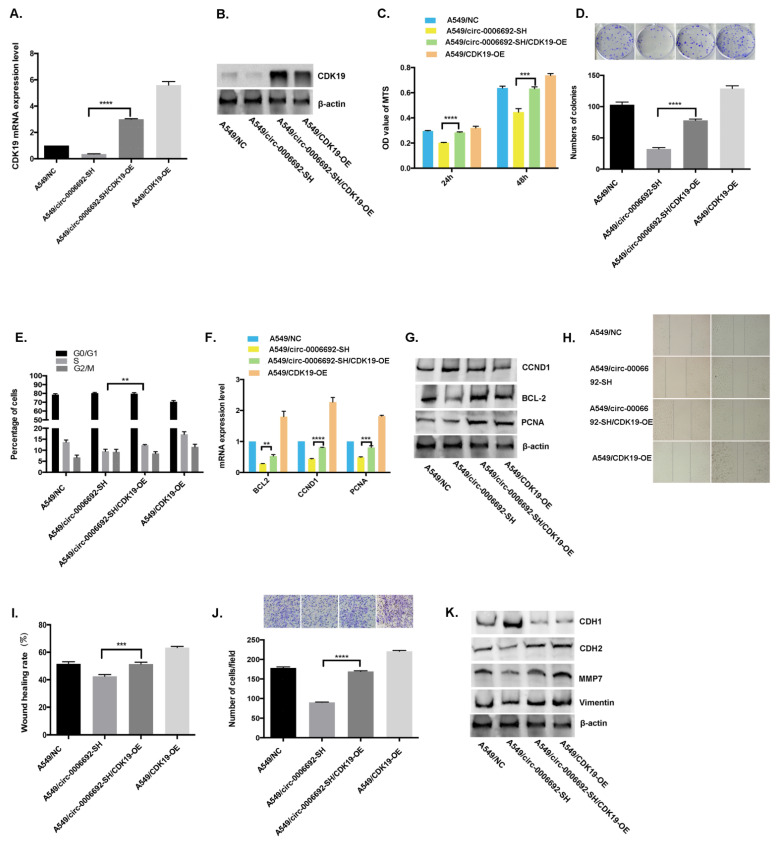
Enhancement of *CDK19* was critical for circ-0006692-mediated malignant behaviors. (**A**,**B**) *CDK19* expression in A549 cells in the rescue assay via RT-PCR and WB. (**C**,**D**) MTS assay and colony formation of A549 cells. (**H**,**I**) Wound healing assay of A549 cells; (**J**) Transwell invasive assay of A549 cells; (**F**,**G**) The expression levels of proliferation-associated-genes of *BCL-2*, *CCND1*, and *PCNA* measured by qRT-PCR and WB; (**K**) The expressions of EMT-associated-genes *MMP7*, *VIMENTIN*, *CDH1*, and *CDH2* demonstrated by Western blotting; (**E**) Cell cycle distribution of A549 cells by FACS; ** *p* < 0.01; *** *p* < 0.001, **** *p* < 0.0001.

**Figure 7 genes-13-00846-f007:**
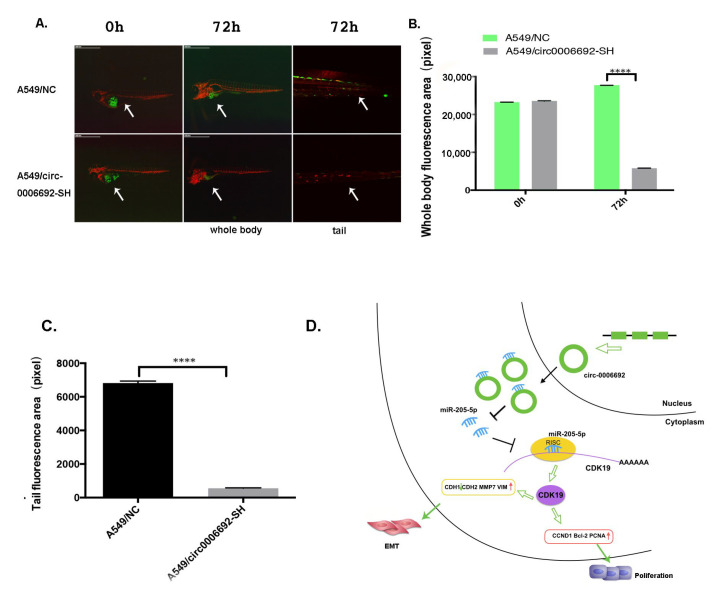
Hsa_Circ_0006692 promoted the growth of lung cancer in zebrafish. (**A**) The fluorescence distribution of lung cancer cells with GFP was observed under stereomicroscope and confocal microscope at different time points; (**B**) The whole body fluorescence area of lung cancer cells at different time points; (**C**) The tail fluorescence area of lung cancer cells at different time points. (**D**) Schematic diagram of the mechanism: hsa_circ_0006692 acts as a miR-205-5p sponge to block its function and up-regulates *CDK19*, forming hsa_circ_0006692/miR-205-5p/CDK19 axis to promote progression and EMT of lung cancer. **** *p* < 0.0001.

**Table 1 genes-13-00846-t001:** Association between hsa_circ_0006692 expression and clinicopathologic features.

Clinicopathologic Feature	Case	Hsa_circ_0006692 lgRQ	*p*-Value
Age			
<60	34	0.4362 ± 0.3259	
≥60	27	0.5196 ± 0.1878	0.8366
Gender			
Male	36	0.3015 ± 0.1748	
Femal	25	0.7202 ± 0.4143	0.3033
Anatomical location			
Central	20	0.1847 ± 0.2176	
Peripheral	41	0.5845 ± 0.2613	0.3706
Tumor size (Maximum diameter)			
≤2	39	0.1849 ± 0.1370	
>2	22	0.9840 ± 0.4819	0.0451 *
Histological classification			
Squamous cell carcinoma	18	0.1296 ± 0.2124	
Adenocarcinoma	43	0.6169 ± 0.2654	0.2662
TNM stage			
I + II	39	0.1849 ± 0.1370	
III + IV	22	0.9840 ± 0.4819	0.0323 *
Lymph node metastasis			
metastasis	33	0.5757 ± 0.3304	
Non-metastasis	28	0.3523 ± 0.1925	0.5791
Invasion of lung membrane			
Not invade	23	0.1554 ± 0.2347	
Invade	38	0.7281 ± 0.3504	0.028 *

* *p* < 0.05.

## Data Availability

The data that support the findings of this study are available from the corresponding author upon reasonable request.

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
