# Peer review of "Hsa_circ_0006692 Promotes Lung Cancer Progression via miR-205-5p/CDK19 Axis"

_genes, 2022, doi:10.3390/genes13050846_

Round 1

Reviewer 1 Report

This manuscript focuses on the role of Hsa_circ_0006692 on lung cancer progression. The authors show that Hsa_circ_0006692 promotes lung cancer progression via miR-205-5p/CDK19 axis. The quality of some experiments might be improved and there are some clarifications required. First, the quality of the data about wound healing is too unclear to distinguish the results. Second, the manuscript needs professional English editing to make the flow better. Finally, how to link Hsa_circ_0006692 and miR-205-5p/CDK19 axis or how to find out the miR-205-5p/CDK19 axis involving the Hsa_circ_0006692 regulation, please explain them in detail.  

Author Response

Reviewer1

The quality of some experiments might be improved and there are some clarifications required. First, the quality of the data about wound healing is too unclear to distinguish the results. Second, the manuscript needs professional English editing to make the flow better. Finally, how to link Hsa_circ_0006692 and miR-205-5p/CDK19 axis or how to find out the miR-205-5p/CDK19 axis involving the Hsa_circ_0006692 regulation, please explain them in detail.。

Response:

  1. The would healing assay was used to evaluate the migration ability after hsa_circ-0006692 was over-expressed on both A549 and H1299 cells. The results of cell migration in the dish were pictured (Figures 2L and N) followed by Image J analysis (Figures 2M and O). The statistical data indicated that compared to the parent cells and mock infection cells, the hsa_circ-0006692 over-expressed A549 and H1299 cells possessed an increased migration ability (P<0.05). In the revision,the quality of pictures has been improved(Fig2L,2N; Fig3L,3N; Fig6H ).

  1. The English has been edited by native English speaker.

  1. To explore if hsa_circ_0006692 was capable of binding to miR-205-5p/CDK19 axis, we performed the following experiments:
  2. RIP analysis, data bases analysis and pull-down test verified that hsa_circ_0006692 was capable of directly interacting with miR-205-5p (Fig 4) .
  3. The qRT-PCR results revealed the changes of hsa_circ_0006692 expression affected the expression levels of miR-205-5p and CDK19 (Fig 4).
  4. Dual luciferase report assay revealed that circ_0006692 had a binding site for miR-205-5p and miR-205-5p also had a binding site for CDK19 (Fig 5).
  5. The qRT-PCR results also revealed that the changes of miR-205-5p affected hsa_circ_0006692 and CDK19 expression levels (Fig 5).

Reviewer 2 Report

Liao J et al. show an interesting article reporting on the biologic role of the hsa_circ_0006692/miR-205-5p/CDK19 408 axis in NSCLC. The issue is relevant since genetic and biologic complexity of NSCLC accounts for its dismal prognosis and resistance to treatments. They found that over-expression of hsa_circ-0006692 was related to malignancy (including some negative clinico-pathological characteristics and EMT) and that it "sponged" miR-205-5p. Furthermore, the hsa_circ_0006692/ miR-205-5p/CDK19 axis increased proliferation, migration, invasion and colony formation. The experiments are well conducted and logically presented. The in vivo experiments (in Zebrafish model) contributed to demonstrate the oncogene potential of hsa_circ-0006692. Discussion and conclusions are consistent with the presented data. Even if interesting and well-conducted the study presentation needs improvements. 

Major comments

I suggest the Authors to improve the informative power of the article by adding quantitative data (numbers) throughout the manuscript. I mean descriptive statistics of relevant expression levels, wound healing rates, number of cells, etc. etc.  In this version, these data are lacking both in manuscript text and abstract.

The Authors should not declare "These results suggested that... was positively related to tumor growth and poor prognosis" (lines 242-243). The prognosis is related to a time-to-outcome analysis not included in their manuscript. "Poor prognosis" should be deleted in both text and title of Figure 1 "hsa_circ-0006692 is up-regulated and associated with poor prognosis of lung cancer".

To demonstrate that Hsa_circ_0006692 displays its oncogene roles through CDK19, the Authors perform a rescue assay in A549/circ_0006692-SH. Why they did not use also the H1299 model? Please, clarify. The scientific manuscript strenght of the manuscript can be improved by adding data from another (this) cell line in this specific experimental context. 

Author Response

Reviewer2:

Liao J et al. show an interesting article reporting on the biologic role of the hsa_circ_0006692/miR-205-5p/CDK19 408 axis in NSCLC. The issue is relevant since genetic and biologic complexity of NSCLC accounts for its dismal prognosis and resistance to treatments. They found that over-expression of hsa_circ-0006692 was related to malignancy (including some negative clinico-pathological characteristics and EMT) and that it "sponged" miR-205-5p. Furthermore, the hsa_circ_0006692/ miR-205-5p/CDK19 axis increased proliferation, migration, invasion and colony formation. The experiments are well conducted and logically presented. The in vivo experiments (in Zebrafish model) contributed to demonstrate the oncogene potential of hsa_circ-0006692. Discussion and conclusions are consistent with the presented data. Even if interesting and well-conducted the study presentation needs improvements.

Major comments

I suggest the Authors to improve the informative power of the article by adding quantitative data (numbers) throughout the manuscript. I mean descriptive statistics of relevant expression levels, wound healing rates, number of cells, etc.  In this version, these data are lacking both in manuscript text and abstract.

Response:

The quantitative data, such as relevant expression levels, wound healing rates, number of cells, etc have been added to revision.

  1. P8 Section 2: Figure 1D showed that the expression level of hsa_circ_0006692 in NSCLC (0.4731 ± 0.1984) was higher than that in adjacent tissues (-0.1608 ± 0.08829, P<0.05).
  2. P8 Section 4: Functionally, the MTS and colony assays revealed that the proliferation of A549 and H1299 cells was 1.62, 1.55 times (MTS) and 1.32 ,1.67 times (colony assays) higher in the hsa_circ_0006692 -OE group compared to the controls (P<0.01, Figures 2B-E), respectively.
  3. P8 Section 4: Furthermore, the wound healing and transwell assays revealed that the migratory abilities and invasive abilities of hsa_circ_0006692 over-expressed A549 and H1299 cells with increased 3.17, 1.51-fold (wound healing assay) and 2.34, 1.57-fold (transwell assay) compared with that in the controls, respectively.
  4. P10 Section 2: Functionally, the MTS (48h) and colony assays revealed that the growth of hsa_circ_0006692 down-regulated A549 and H1299 cells decreased by 46.7%, 25.7% (MTS) and 28.2%, 54.9% (colony assay) compared to the controls (P<0.001, Figures 3B-E), respectively.
  5. P17 Section 2: The results (Figures 7A-B) of in vivo zebrafish assay with stereomicroscope and confocal microscope at different time points to trace the fluorescence distribution of lung cancer cells with GFP (Green fluorescent protein) showed that the GFP-A549/circ-0006692-SH cells were 20% lower (P<001) than that of A549/NC in zebrafish at 72 hours, indicating that knockdown of circ-0006692 inhibited the proliferation of lung cancer cells in vivo as observed in vitro. In addition, the tail GFP area of A549/circ-0006692-SH was 8% lower (P<0.001) than that of A549/NC at 72 hours, indicating that knockdown of hsa_circ-0006692 suppressed the migration of lung cancer cells (Figures 7C).

The Authors should not declare "These results suggested that... was positively related to tumor growth and poor prognosis" (lines 242-243). The prognosis is related to a time-to-outcome analysis not included in their manuscript. "Poor prognosis" should be deleted in both text and title of Figure 1 "hsa_circ-0006692 is up-regulated and associated with poor prognosis of lung cancer".

Response:

According to reviewer suggestion, the line 242-243 has been changed to “these results suggested that... was significantly related to clinicopathologic features,such as tumor size,TNM and invasion of lung basal membrane”, and the “poor prognosis” has been deleted. Similarly, both text and title of Figure 1 "hsa_circ-0006692 is up-regulated and associated with poor prognosis of lung cancer" have been taken out.

To demonstrate that Hsa_circ_0006692 displays its oncogene roles through CDK19, the Authors perform a rescue assay in A549/circ_0006692-SH. Why they did not use also the H1299 model? Please, clarify. The scientific manuscript strenght of the ,specific experimental context.

Response:

This study is basic research for deeply exploring the interactions among three molecules and defining the hsa_circ-0006692 / miR-205-5p /CDK19 axis using modern molecular assays, such as over-expression, knocked-down, rescue assay, pull-down assays, etc. In most of experiments, we used two cell lines, and the data and figures are almost full for readers to digest. In some experiments, such as the rescue assay, due to the overwhelming workload and short of some reagents, we performed only in A549, and the results are clear-cut.    
